# Identification of Molecular Fragments in Equilibrium with Polysiloxane Ultrasmall Nanoparticles

**DOI:** 10.3390/nano12050738

**Published:** 2022-02-22

**Authors:** Paul Rocchi, Lucie Labied, Tristan Doussineau, Michel Julien, Barbara Giroud, Emmanuelle Vulliet, Jérôme Randon, Olivier Tillement, Agnès Hagège, François Lux

**Affiliations:** 1Institut Lumière Matière, Université Claude Bernard Lyon 1, CNRS UMR 5306, 69622 Villeurbanne, France; rocchi@nhtheraguix.com (P.R.); lucie.labied@isa-lyon.fr (L.L.); olivier.tillement@univ-lyon1.fr (O.T.); 2NH TherAguix S.A, 29 Chemin du Vieux Chêne, 38240 Meylan, France; doussineau@nhtheraguix.com (T.D.); julien@nhtheraguix.com (M.J.); 3Institut des Sciences Analytiques, CNRS, Université Claude Bernard Lyon 1, Université de Lyon, UMR 5280, 69100 Villeurbanne, France; barbara.giroud@isa-lyon.fr (B.G.); emmanuelle.vulliet@isa-lyon.fr (E.V.); jerome.randon@univ-lyon1.fr (J.R.); agnes.hagege@univ-lyon1.fr (A.H.); 4Institut Universitaire de France (IUF), 75000 Paris, France

**Keywords:** ultrasmall nanoparticles, theranostic, polysiloxane, hyphenated high-performance liquid chromatography, inductively coupled plasma mass spectrometry

## Abstract

During recent decades, ultrasmall inorganic nanoparticles have attracted considerable interest due to their favorable biodistribution, pharmacokinetics and theranostic properties. In particular, AGuIX nanoparticles made of polysiloxane and gadolinium chelates were successfully translated to the clinics. In an aqueous medium, these nanoparticles are in dynamic equilibrium with polysiloxane fragments due to the hydrolysis of Si–O–Si bonds. Thanks to high-performance liquid chromatography coupled with electrospray ionization mass spectrometry, all these fragments were separated and identified.

## 1. Introduction

Due to their capacity to integrate different functionalities, nanoparticles have attracted considerable interest for biomedical applications, in particular for theranostic ones (a combination of diagnostic and therapeutic), a concept introduced by Funkhouser in 1998 [1,2]. In oncology, nanoparticles are of particular interest since they can be passively accumulated in tumors after intravenous administration through the enhanced permeability and retention (EPR) effect [3]. However, despite intense preclinical research [4,5] and important funding during the last few decades, only a limited number of nanoparticles have reached clinical trials and even less to the market [6,7]. Several explanations can be given to explain this gap: (i) difficulty to scale-up the production and lack of process robustness; (ii) lack of detailed product physicochemical characterization to convince regulatory offices on these new types of pharmaceutical products; (iii) unexpected toxicity; (iv) detrimental discrepancies between promising preclinical results and clinical ones; or (v) long, uncertain, and expensive clinical development leading to a high risk of failure before commercialization [8,9,10]. Nanoparticle-based pharmaceuticals are complex products for which regulatory guidelines are still not properly defined. Nanomedicines require the implementation and the development of new and specific analytical techniques and methods that have many hurdles in the regulatory pathway up to their approval.

Among nanoparticles, ultrasmall inorganic nanoparticles (with a hydrodynamic diameter of <10 nm) were specifically developed thanks to their unique advantages [11,12]. Indeed, they can combine the benefits of including metals displaying interesting properties (such as magnetic properties for magnetic resonance imaging or high Z for radiosensitization) and fast renal elimination avoiding long-term retention of metals in the body, often associated with toxicity. In addition, even if their resident time in the blood circulation is lesser than for larger nanoparticles, better tumoral penetration can be achieved [3]. Due to these interesting features, two ultrasmall nanoparticles were accepted in clinical trials during the last decade: (i) Cornell dots made of silica core embedding an optical imaging agent and displaying PEG functions, ^124^I agent and cRGD [13], and (ii) AGuIX made of a polysiloxane (silica derivative) matrix onto which macrocyclic gadolinium chelates (DOTAGA(Gd)) are covalently grafted (See Figure 1A,B) [14]. A phase 1b clinical trial on the treatment of brain metastases by whole-brain radiation therapy using AGuIX as a radiosensitizer (NanoRad) was recently completed demonstrating the very good safety profile of the product, as well as first hints at efficacy [15,16]. Based on these encouraging results, AGuIX is now in phase 2 clinical trials for this indication and for others, such as lung and pancreatic cancers.

Due to their high surface-to-volume ratio, ultrasmall silica-based nanoparticles are more susceptible to be biodegraded [17]. The degradation mechanism can be described by a dissolution of the silica-based matrix through the hydrolysis of Si–O–Si bonds over time and yielding (poly)silicic acid and silanols molecules that are finally fragments of the product. This kind of biodegradation has already been shown on AGuIX nanoparticles using high liquid performance chromatography (HPLC) coupled to electrospray ionization mass spectrometry (ESI-MS) [18] or Taylor dispersion analysis (TDA) coupled to inductively coupled plasma mass spectrometry (ICP-MS) but these methods were not capable of providing clear identification for all molecular fragments in equilibrium [19]. In these studies, assessment of biodegradability was mainly performed by monitoring the size of the nanoparticle over time and molecular fragments were proposed after almost complete hydrolysis of the polysiloxane matrix. Due to hydrolysis reactions, polysiloxane nanoparticles are always in equilibrium with molecular fragments in an aqueous medium. The aim of this paper is to clearly identify the exact nature of these fragments for the AGuIX nanoparticles at a relevant concentration for in vivo applications.

## 2. Materials and Methods

### 2.1. Materials

Acetonitrile (CH_3_CN, ACN, >99.9%) was purchased from Sigma Aldrich (Saint-Quentin-Fallavier, France). Trifluoroacetic acid (TFA) and Formic acid (AF) at LC/MS grade purity were purchased from Fischer Scientific (Waltham, MA, USA). The starting USNPs AGuIX (gadolinium-chelated polysiloxane nanoparticles) were provided by NH TherAguix (Meylan, France) as lyophilized powder and reconstituted in ultrapure water. Only Milli-Q water (ρ > 18 MΩ·cm) was used for the aqueous solution preparation.

### 2.2. Instrumentation

HPLC-UV/Vis system: A Shimadzu Prominence series UFLC system with a CBM-20A controller bus module, an LC-20 AD liquid chromatograph, a CTO-20A column oven and an SPD-20A UV-visible detector. UV-visible absorption was measured at 295 nm.

HPLC-ICP/MS system: A Nexion 2000B (Perkin-Elmer, Villebon Sur Yvette, France), coupled with a Flexar LC system (Perkin-Elmer). Gd signal was monitored at *m*/*z* 152. Syngistix software version 2.3 was used to control the ICP-MS. The Gd signal was acquired through Empower software version 7.3.

HPLC-ESI/MS system: The ESI/MS measurements were carried out on a triple quadrupole spectrometer Xevo TQ-S (Waters, Milford, MA, USA), coupled with UHPLC chain Acquity H-Class (Waters). The analyses were performed on positive mode (ESI+) and detection SCAN mode set on 300–1500 uma range.

### 2.3. Elemental Analysis

The measure of gadolinium and silicium content in the AGuIX product was conducted by the company Quality Assistance (Thuin, Belgium). The measurements were performed through ICP-MS Agilent 7900 equipped with Masshunter software. The measure of carbon and nitrogen content was conducted by the Isotope and Organic Laboratory of Institut des Sciences Analytiques (ISA, Lyon, France) through device designed by the laboratory. For the carbon analysis, total combustion of the AGuIX at 1050 °C under a stream of oxygen is performed. The carbon in the sample is transformed into carbon dioxide then quantified by specific infrared CO_2_ detectors. For the nitrogen, total combustion of AGuIX sample at 1050 °C is performed under a stream of helium and oxygen. The nitrogen in AGuIX is transformed into various nitrogen oxides reduced to molecular nitrogen. Carbon dioxide and water from combustion are trapped on ascarite and magnesium perchlorate. Nitrogen is quantified by a thermal conductivity detector.

### 2.4. Separation of Both Nanoparticles and Fragments in a Single Run

The AGuIX sample composition was studied through HPLC using conditions described in C. Truillet et al. [16] The separation was performed using C4 reverse phase column (Jupiter^®^, 5 µm, 300 Å, 150 × 4.6 mm) at constant flow rate of 1 mL·min^−1^. The gradient initial mobile phase was 95% solvent A—5% solvent B (A = H_2_O/ACN/TFA: 98.9/1/0.1 *v*/*v*/*v*, B = ACN/H_2_O/TFA: 89.9/10/0.1 *v*/*v*/*v*) and was held for 5 min. After this isocratic step, nanoparticles were eluted by a gradient developed from 5% to 90% of solvent B in solvent A over 15 min. The concentration of solvent B was maintained over 5 min.

The sample was analyzed using both UV-Vis detection and ICP-MS detection. For the ICP-MS analysis, O_2_ was added to the gas injection mixture in order to facilitate complete combustion of the increasing amount of ACN in the plasma source. Moreover, after the elution of the nanoparticle peak, the LC route was switched to the waste to avoid excess ACN entering the ICP. Operating conditions used for this analysis by ICP-MS were: nebulizer gas flow rate, 0.77 L/min; O_2_ AMS Gas flow, 0.07 L.min^−1^; plasma gas flow rate, 15 L.min^−1^; auxiliary gas flow rate, 1.2 L.min^−1^; radiofrequency power, 1600 W for the plasma. All other parameters were tuned to maximize the Gd signal at *m*/*z* = 152.

### 2.5. Fragment Separations and Identification by HPLC Coupled to Different Detectors

The fragment separation was achieved by HPLC using C4 reverse phase column (Jupiter^®^, 5 µm, 300 Å, 150 × 2 mm). The measurements were performed on isocratic mode using the following phase composition H_2_O/ACN/AF (98.9%/1%/0.1%) at 0.2 mL·min^−1^ flow rate. After each nanoparticle injection, the column was flushed with H_2_O/ACN/AF (19.9/80/0.1 *v*/*v*/*v*) solution for 10 min, to elute the nanoparticles. The fragments were analyzed using the three different detectors. The operating condition used for the ICP-MS system were: nebulizer gas flow rate, 0.84 L·min^−1^; plasma gas flow rate, 15 L·min^−1^; auxiliary gas flow rate, 1.2 L·min^−1^; radiofrequency power, 1600 W for the plasma. The operating conditions used for the ESI/MS system were: capillary tension, 3.2 kV; source temperature, 150 °C; desolvation temperature, 550 °C; N_2_ desolvation gas flow, 900 L·h^−1^; N_2_ nebulizing gas flow, 150 L·h^−1^.

The identification and the drawing of each fragment were assisted by the use of the MarvinSketch software (ChemAxon, https://www.chemaxon.com, accessed on 16 February 2022).

## 3. Results and Discussion

### 3.1. Synthesis and Purification of AGuIX Nanoparticles

The representative AGuIX^®^ batch studied in this article was prepared through a five-step process consisting of three chemical transformation steps, a purification step and a lyophilization step: (i) formation of a gadolinium oxide core by addition of sodium hydroxide on gadolinium chloride in diethylene glycol (DEG); (ii) growth of a polysiloxane shell by addition of a mixture of TEOS (tetraethoxysilane) and APTES (aminopropyltriethoxysilane); (iii) functionalization of the amino functions issued from APTES by DOTAGA anhydride (2,2′,2″-(10-(2,6-dioxotetrahydro-2H-pyran-3-yl)-1,4,7,10-tetraazacyclododecane-1,4,7-triyl)triacetic acid); (iv) after precipitation in acetone and transfer to water, the nanoparticles are purified through tangential flow filtration during which a top-down process occurs consisting of dissolution of the gadolinium oxide cores due to chelation of gadolinium by DOTAGA followed by fragmentation of polysiloxane shells, rearrangement of these fragments and chelation of released gadolinium by DOTAGA (see Appendix A) [20,21]. Tangential flow filtration using membranes of NMWCO of 5kDa aims at removing small molecules comprising salts, residual solvents, any residual molecular species and the smallest polysiloxane species. They are finally freeze-dried for easy handling. After dispersion in water, the nanoparticles display a hydrodynamic diameter of 2.7 nm (see Appendix A), and almost all of the DOTAGA chelate a gadolinium ion (~1–2% of free chelates) (see Appendix A).

### 3.2. Separating Nanoparticles from Fragments by HPLC Coupled to Different Detection Methods

Despite a high purification factor applied by tangential flow filtration, fragments issued from the polysiloxane matrix are always detected by HPLC due to the equilibrium between them and the nanoparticles after hydrolysis of the Si–O–Si bonds (see Figure 1C.). Fragments are detected between 2 and 4 min and nanoparticles between 9 and 14 min on a C4 reverse-phase column using both UV/VIS and ICP-MS detectors.

The HPLC conditions have further been modified to match with the ESI-MS detector requirements. Consequently, the flow rate was lowered by reducing the reverse phase column diameter from 4.6 mm to 2 mm, and TFA was replaced by AF in the HPLC phases to enhance the ESI-MS signal of the fragments. Those conditions were applied to study the separation of the different fragments using three different types of detection (i.e., ICP/MS, ESI/MS and UV). ESI/MS detection permits us to distinguish seven peaks while ICP/MS and UV detection at 295 nm are limited to four or five peaks. This discrepancy led us to consider that the two first peaks are associated with polysiloxane fragments that do not contain either gadolinium (detected by ICP/MS) or DOTAGA (detected by UV detection at 295 nm) (Figure 2).

### 3.3. Identification of the Different Species by ESI/MS

As previously performed by Hu et al. for carbon nanoparticles [22], the different peaks separated by HPLC were fully studied by ESI/MS (see Figure 3) and the identification of the fragments was proposed in Table 1 using notable gadolinium isotopic patterns and water hydration to determine the charge states of the compounds. As stated previously, thanks to ICP-MS and UV detections, the two first peaks are fragments issued from polysiloxane without DOTAGA(Gd) sub-structures. For the first HPLC peak, three co-eluted components were identified (see Appendix A). For each, the mass is obtained involving a methanoate anion present in the eluent. The *m*/*z* signal at 303.0 (z = 1) is related to a fragment corresponding to two silicon atoms issued from APTES. A signal corresponding to an *m*/*z* of 423.0 is associated with four silicon atoms coming from APTES for two of them and TEOS for the two others. Interestingly, the signal at an *m*/*z* of 441.1 corresponds to the same species with the hydrolysis of one Si–O–Si bond. For the less defined HPLC peak 2 (see Appendix A) at a time of 2.19 min, larger polysiloxane fragments are observed. The intense signal at an *m*/*z* of 301.52 corresponds to a fragment with six silicon atoms including three issued from APTES and three issued from TEOS. The signal at an *m*/*z* of 394.5 corresponds to the same entity with the addition of one Si(OH)_3_ issued from TEOS. The signal at an *m*/*z* of 397.5 is obtained by the addition of another Si(OH)_3_ and the hydrolysis of two Si–O–Si bonds. Higher retention is observed for compounds containing DOTAGA(Gd) entities that are present in HPLC peaks 3 to 7. For HPLC peak 3, only one main entity (see Appendix A) is observed with different charge and hydrolysis states. The *m*/*z* signal of 391.4 (z = 3) corresponds to a DOTAGA(Gd) coupled to an aminopropyl with six silicon atoms issued from two APTES and four TEOS. Signals at an *m*/*z* of 397.4 and 403.6 are the products resulting from one and two hydrolysis of the Si–O–Si bonds. The *m*/*z* of 586.5 and *m*/*z* of 595.5 are associated with an *m*/*z* of 391.4 and 397.4, respectively, but for z = 2. The mass spectrum of HPLC peak 4 is relatively similar to that of HPLC peak 3. It corresponds also to one main species with different charge and hydrolysis states (see Appendix A). The signal at an *m*/*z* of 358.0 (z = 3) corresponds to DOTAGA(Gd) coupled to an aminopropyl with five silicon atoms issued from two APTES and three TEOS. Hydrolysis of one or two Si–O–Si bonds leads to the species at an *m*/*z* of 364.0 and 370.0, respectively. For the three species, another charge state is also detected (z = 2) at an *m*/*z* of 536.5, 545.6 and 554.6 corresponding to the precedent species either not hydrolyzed or with the hydrolysis of one or two Si–O–Si bonds, respectively. For HPLC peak 5, a compound containing two DOTAGA(Gd) species is observed at an *m*/*z* of 506.8 (see Figure 4). It contains also nine silicon atoms issued from three APTES and six TEOS. The species at an *m*/*z* of 515.54 are related to the precedent ones with the hydrolysis of two Si–O–Si bonds. Interestingly other detected species contain only one DOTAGA(Gd) molecule. This can be explained by the cleaving hydrolysis of the Si–O–Si in the ESI-MS source leading to two species detected at an *m*/*z* of 486.0 and 536.5. The first one is composed of one DOTAGA(Gd) and four silicon atoms issued from three TEOS and one APTES. The hydrolysis of one Si–O–Si bond leads to a species detected at an *m*/*z* = 495.0 for z = 2 and 330.4 for z = 3. For the second one, a composition of one DOTAGA(Gd) and five silicon atoms issued from three TEOS and two APTES is obtained. Hydrolysis of one Si–O–Si bond leads to *m*/*z* of 545.6 for z = 2 and 364.0 for z = 3. For HPLC peak 6 also, the main species corresponds to two DOTAGA(Gd) and twenty silicon atoms issued from nine APTES and eleven TEOS (see Appendix A). As for HPLC peak 5, cleaving hydrolysis of a Si–O–Si bond occurs in the ESI-MS source and leads to two different species with different charge states. The first one corresponds to one DOTAGA(Gd) associated with nine silicon atoms issued from four APTES and five TEOS and is observed at an *m*/*z* of 367.3 for z = 4 and of 733.6 for z = 2. By hydrolysis or condensation, the species at an *m*/*z* of 724.6, 742.6, 751.6 or 760.6 can be obtained (see Figure 5). The second one corresponds to one DOTAGA(Gd) associated with eleven silicon atoms issued from five APTES and six TEOS and corresponds to masses of 425.5 (z = 4) and 850.1 (z = 2). Finally, HPLC peak 7 is attributed to DOTAGA(Gd) without polysiloxane fragments (see Appendix A). This was verified by an LC-ESI-MS analysis of a control sample of DOTAGA(Gd) (see Appendix A).

Altogether, these results show the biodegradability of the nanoparticle and its equilibrium with smaller fragments issued from the cleaving hydrolysis of Si–O–Si functions. It has already been shown that a large part of these fragments is eliminated very rapidly, as fragments under 8 kDa are mainly observed five minutes after intravenous administration in rats, and their concentration decreases a lot after this timepoint [23]. One of the limitations of the study is the use of an aqueous medium that cannot completely mimic the more complex interactions that may arrive in a biological medium but a precedent biodegradation study performed in phosphate buffer and serum has shown a very similar electrophoretic profile for the molecular fragments in both media [19]. Moreover, AGuIX presents very limited interactions with proteins, such as human serum albumin [24], confirmed by their very fast renal elimination [15,16]. Altogether, these results indicate that the fragments identified in this study will certainly also be representative of the biological medium.

## 4. Conclusions

Full identification of silanol molecular species issued from AGuIX partial dissolution in an aqueous medium was performed. These molecular species are fragments of the ultrasmall AGuIX nanoparticle that are in dynamic equilibrium with the nanoparticles at a given concentration. They display from one to two DOTAGA (Gd) entities. These fragments are of no safety concern due to their silane composition and also their rapid elimination through the kidneys [23]. The approach developed in this study can be applied to different ultrasmall nanoparticles that will certainly be mainly developed in the future due to their interesting pharmacokinetic features.

## Figures and Tables

**Figure 1 nanomaterials-12-00738-f001:**
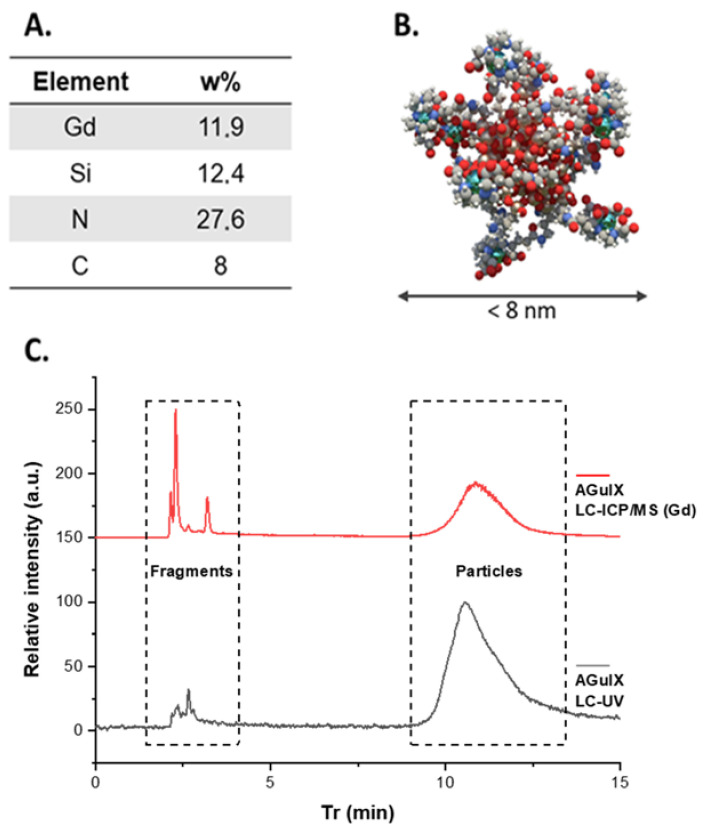
(**A**) Elemental analysis of AGuIX (**B**) Schematic representation of AGuIX NPs (gadolinium atoms in green are chelated in DOTAGA ligands grafted to polysiloxane matrix). (**C**) LC-ICP/MS chromatogram of AGuIX (5 µL, 10 g/L) recorded at Gd 152 channel (red). LC-UV chromatogram of AGuIX (5 µL, 10 g/L) recorded at λ = 295 nm (black).

**Figure 2 nanomaterials-12-00738-f002:**
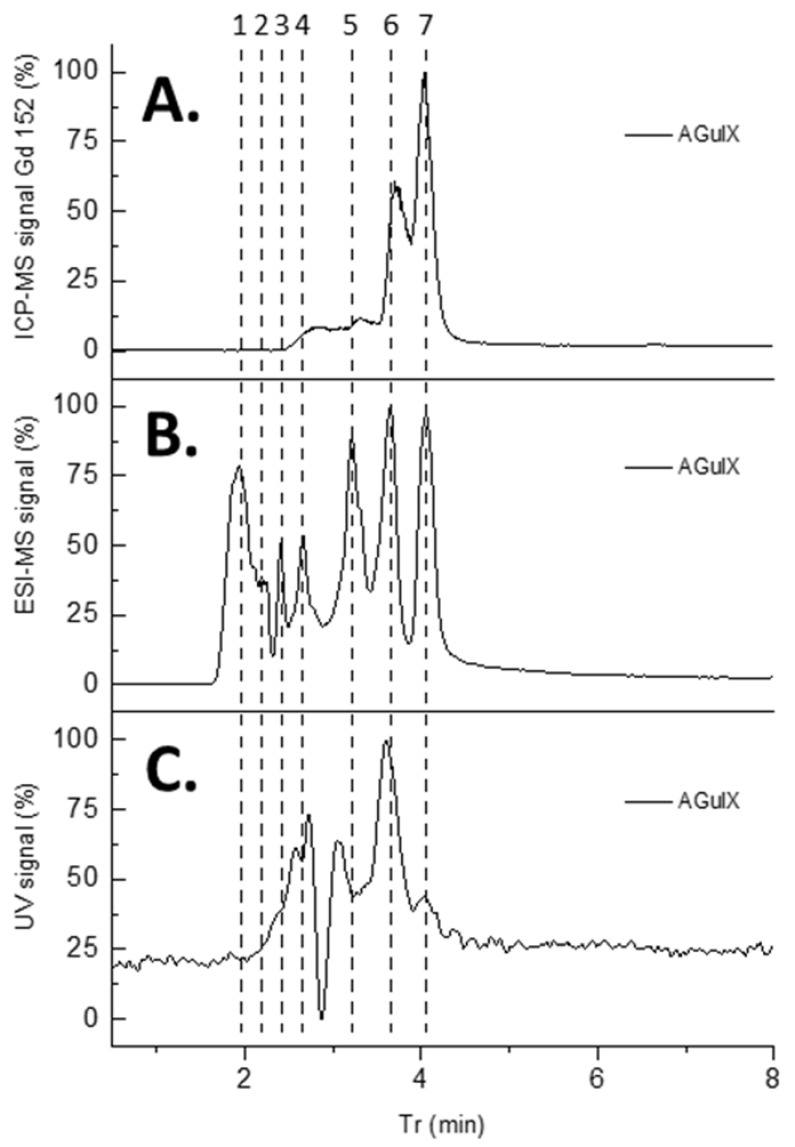
(**A**) LC-ICP/MS chromatogram of AGuIX’s fragments (3 µL, 1 g/L) recorded at Gd 152 channel (**B**) LC-ESI/MS chromatogram of AGuIX’s fragments (3 µL, 1 g/L) set on mass range 300–1500 uma. (**C**) LC-UV chromatogram of AGuIX’s fragments (10 µL, 1 g/L) recorded at λ = 295 nm.

**Figure 3 nanomaterials-12-00738-f003:**
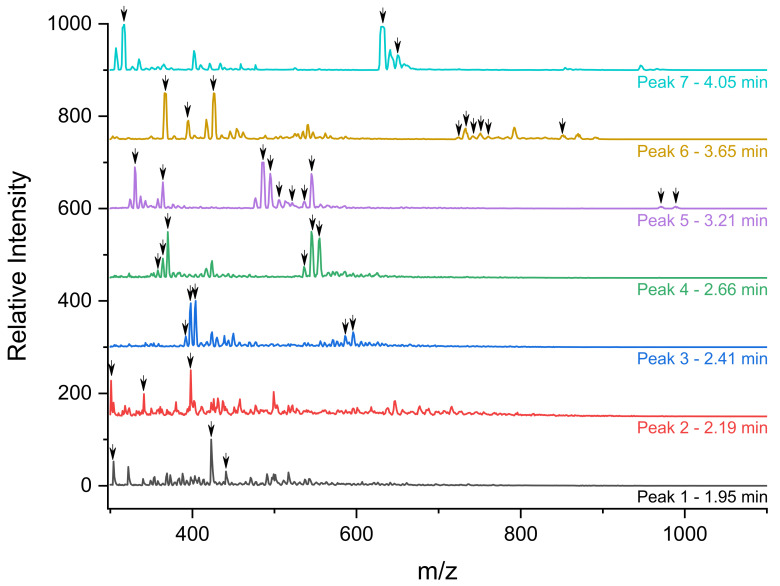
MS spectra of each peak identified (1–7) in the LC-ESI/MS AGuIX’s fragments chromatogram. All identified *m*/*z* are pointed with a black arrow.

**Figure 4 nanomaterials-12-00738-f004:**
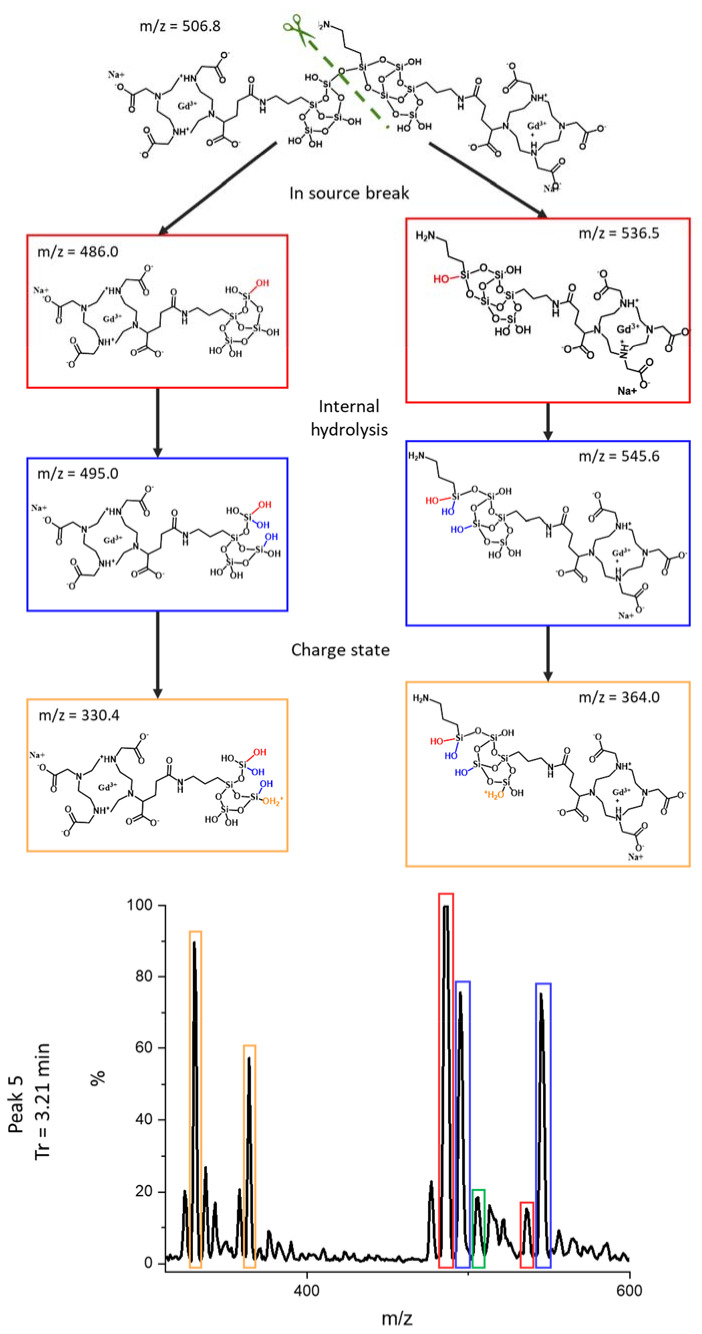
Stucture of the main chemical compound detected on peak 5 (Tr = 3.21 min) and of the related species issued from the hydrolysis of the Si–O–Si bond.

**Figure 5 nanomaterials-12-00738-f005:**
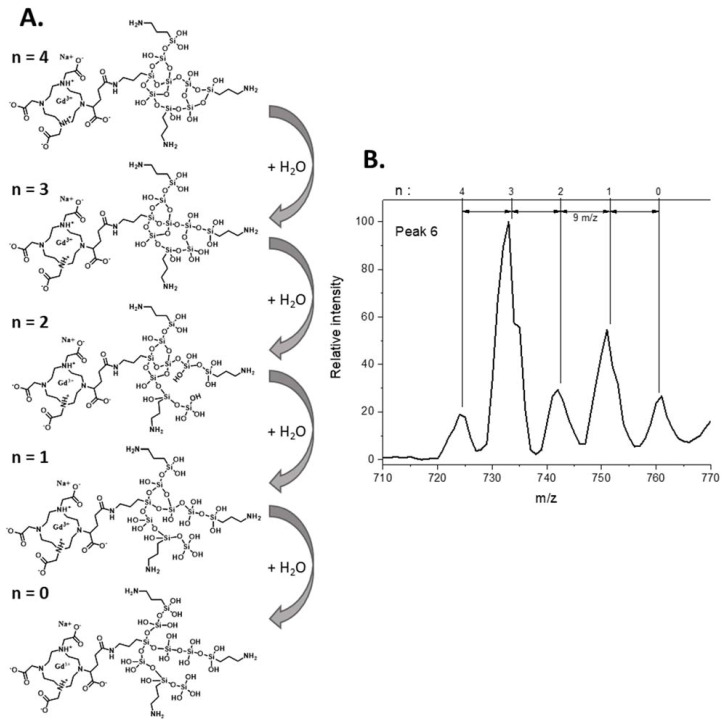
As illustrated by study of peak 6 fragment, different hydrolysis and condensation reaction can occur leading to addition or removal of H_2_O to the structure. (**A**) Scheme of successive siloxane bridge hydrolysis going from *n* = 4 to *n* = 0 on fragment detected at *m*/*z* = 724.6. (**B**) Zoom on the related peak 6 MS spectra area (710–770 *m*/*z*) where the successive *m*/*z* of hydrolyzed fragment can be found.

**Table 1 nanomaterials-12-00738-t001:** Proposed fragments formula for *m*/*z* observed in MS spectra of peak 1 to 7, with associated retention time from 1.95 to 4.05 min.

Time (min)	*m*/*z*	Species	Peak
1.95	303.0	C_6_H_22_N_2_O_5_Si_2_^2+^ + CHO_2_^−^	1
	423.0	C_6_H_22_N_2_O_9_Si_4_^2+^ + CHO_2_^−^	
	441.1	C_6_H_24_N_2_O_10_Si_4_^2+^ + CHO_2_^−^	
2.19	301.5	C_9_H_32_N_3_O_13_Si_6_^3+^ + CHO_2_^−^	2
	340.5	C_9_H_34_N_3_O_16_Si_7_^3+^ + CHO_2_^−^	
	397.5	C_9_H_40_N_3_O_21_Si_8_^3+^ + CHO_2_^−^	
2.41	391.7	C_28_H_56_GdN_7_O_21_Si_6_^2+^ + Na^+^	3
	397.7	C_28_H_58_GdN_7_O_22_Si_6_^2+^ + Na^+^	
	403.7	C_28_H_60_GdN_7_O_23_Si_6_^2+^ + Na^+^	
	587.1	C_28_H_55_GdN_7_O_21_Si_6_^+^ + Na^+^	
	596.1	C_28_H_57_GdN_7_O_22_Si_6_^+^ + Na^+^	
2.66	358.0	C_25_H_49_GdN_6_O_20_Si_5_^2+^ + Na^+^	4
	364.0	C_25_H_51_GdN_6_O_21_Si_5_^2+^ + Na^+^	
	370.0	C_25_H_53_GdN_6_O_22_Si_5_^2+^ + Na^+^	
	536.5	C_25_H_48_GdN_6_O_20_Si_5_^+^ + Na^+^	
	545.6	C_25_H_50_GdN_6_O_21_Si_5_^+^ + Na^+^	
	554.6	C_25_H_52_GdN_6_O_22_Si_5_^+^ + Na^+^	
3.21	330.4	C_22_H_44_GdN_5_O_20_Si_4_^2+^ + Na^+^	5
	364.0	C_25_H_51_GdN_6_O_21_Si_5_^2+^ + Na^+^	
	486.0	C_22_H_41_GdN_5_O_19_Si_4_^+^ + Na^+^	
	495.0	C_22_H_43_GdN_5_O_20_Si_4_^+^ + Na^+^	
	506.8	C_47_H_87_Gd_2_N_11_O_38_Si_9_^2+^ +2 Na^+^	
	515.5	C_47_H_91_Gd_2_N_11_O_40_Si_9_^2+^ + 2 Na^+^	
	536.5	C_25_H_48_GdN_6_O_20_Si_5_^+^ + Na^+^	
	545.6	C_25_H_50_GdN_6_O_21_Si_5_^+^ + Na^+^	
	971.1	C_22_H_40_GdN_5_O_19_Si_4_ + Na^+^	
	989.1	C_22_H_42_GdN_5_O_20_Si_4_ + Na^+^	
3.65	367.3	C_31_H_72_GdN_8_O_30_Si_9_^3+^ + Na^+^	6
	394.2	C_65_H_157_Gd_2_N_17_O_66_Si_20_^6+^ + 2 Na^+^	
	425.5	C_34_H_87_GdN_9_O_37_Si_11_^3+^ + Na^+^	
	724.6	C_31_H_68_GdN_8_O_29_Si_9_^+^ + Na^+^	
	733.6	C_31_H_70_GdN_8_O_30_Si_9_^+^ + Na^+^	
	742.6	C_31_H_72_GdN_8_O_31_Si_9_^+^ + Na^+^	
	751.6	C_31_H_74_GdN_8_O_32_Si_9_^+^ + Na^+^	
	760.6	C_31_H_76_GdN_8_O_33_Si_9_^+^ + Na^+^	
	850.1	C_34_H_85_GdN_9_O_37_Si_11_^+^ + Na^+^	
4.05	316.6	C_19_H_31_GdN_4_O_10_^2+^	7
	632.1	C_19_H_30_GdN_4_O_10_^+^	
	650.1	C_19_H_30_GdN_4_O_10_^+^ + H_2_O	

## Data Availability

The data presented in this study are available on request from the corresponding author.

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
