# Peer review of "Identification of Molecular Fragments in Equilibrium with Polysiloxane Ultrasmall Nanoparticles"

_nanomaterials, 2022, doi:10.3390/nano12050738_

Round 1

Reviewer 1 Report

The rationale and goal are clearly indicated in the introduction. Experiments are described in detailed manner and results are clearly presented.

Overall the manuscript is interesting and I suggest the publication with minor revisions listed below.

1) In the introduction authors should mention the obtained results to state in a clearer manner  the practical  advantage of their approach compared with those  already published ( study the nanoparticles biodegradability by monitoring their size and that analyze the fragment at the end of  the polysiloxane matrix complete hydrolysis).

2) In my opinion the detailed description as well as the scheme for synthesis of AGuIX should be reported in the main text. 

Reviewer 2 Report

This manuscript constructed the ultrasmall inorganic AGuIX nanoparticles, which made of polysiloxane and gadolinium chelates. In aqueous medium, AGuIX nanoparticles are in dynamic equilibrium with polysiloxane fragments due to hydrolysis of Si-O-Si bonds. In this work, the fragments have been separated and identified by using high performance liquid chromatography coupled to electrospray ionization mass spectrometry. The separation and identify of the fragments parts is relatively complete. However, the lack of some key data about the successful preparation of AGuIX nanoparticles prevents me to recommend this work published in Nanomaterials directly. I suggest the authors should pay more patient to revise their paper and perform more experiments to convince their results.

  1. The authors need to prove successful preparation of AGuIX nanoparticles, such as the available functional groups of AGuIX nanoparticles should be validated by Fourier Transform Infrared spectroscopy; the morphological features of the AGuIX nanoparticles should be studied in a Transmission Electron Microscope or Scanning Electron Microscope; the zeta-potential and dispersity index of the nanoparticles should be detected by DLS.
  2. Data in Figure 1A. The authors need to provide the method about the calculation method of the w% of the element in AGuIX nanoparticles. Also, it would be better to provide the EDS mapping pictures of the nanoparticles.
  3. The author's detection condition is an aqueous solution. If it is under physiological conditions (there will be a lot of enzymes, small molecules, etc.), will AGuIX nanomaterials be hydrolyzed in the same way?
  4. The quality about Figure 1 C and chemical structures in Figures 4 and 5 need to be improved. Also, “Si-O-SI” in Figure 4 should be changed to“Si-O-Si”.
  5. In the text, some great related references about the properties and benefits of nanomaterials can be cited, such as: Zhao L, Xing Y, Wang R, et al. Self-assembled nanomaterials for enhanced phototherapy of cancer[J]. ACS Applied Bio Materials, 2019, 3(1): 86-106; Zhao L, Liu H, Xing Y, et al. Tumor microenvironment-specific functional nanomaterials for biomedical applications[J]. Journal of Biomedical Nanotechnology, 2020, 16(9): 1325-1358.

Round 2

Reviewer 2 Report

Based on the reviewer's suggestion, the author has made extensive revisions and I recommend it for publication